# Impact of Selected Geriatric Syndromes on the Quality of Life in the Population Aged 60 and Older

**DOI:** 10.3390/healthcare9060657

**Published:** 2021-05-31

**Authors:** Jitka Doležalová, Valérie Tóthová, Jan Neugebauer, Petr Sadílek

**Affiliations:** 1Institute of Nursing, Midwifery and Emergency Care, Faculty of Health and Social Sciences of University of South Bohemia, 37001 České Budějovice, Czech Republic; tothova@zsf.jcu.cz (V.T.); neugebauer@zsf.jcu.cz (J.N.); 2Medical Information Centre, 11000 Prague, Czech Republic; sadilek@help-lic.cz

**Keywords:** nursing assessment, quality of life, geriatric syndromes, Rapid Geriatric Assessment, frail, SARC-F, Rapid Cognitive Screen, The Simplified Nutritional Appetite Questionnaire, geriatric nursing

## Abstract

Background: Geriatric syndromes represent a critical domain in the population more than 60 years old. Basic syndromes include frailty, sarcopenia, loss of body mass, and a mild cognitive disorder. These are significant problems which can affect the quality of life. In our study, the Rapid Geriatric Assessment (RGA) tool was used to assess the geriatric syndromes, and the WHOQOL-BREF was used to assess the quality of life to survey 498 respondents from a population aged 60 and older. In all the assessments, the distribution of variables was tested, a nonnormal distribution of variables was identified, and subsequently, nonparametric tests were performed to identify the differences between groups. The study showed that the domain of physical health and the psychological domain were most affected. The results have shown that individual geriatric syndromes affect certain domains of the quality of life of the population above 60 with various intensity.

## 1. Introduction

The World Health Organization (WHO) defines the quality of life as “an individual’s perception of their position in life in the context of the culture and value systems in relation to their goals, expectations, standards and concerns” [1]. It is a multidimensional and holistic entity, which can be assessed in several disciplines [2]. However, no exact definition of the quality of life can be determined since it is a highly subjective concept [3]. The topic involves psychological, physical, and social aspects of life [4]. The older adult’s quality of life is defined by feeling well, performing daily activities, and being independent [5]. Older adult individuals who grow old successfully are satisfied with their past and present, which is associated with the subjective perception of their wellness and adaptation to the current environment [6].

Geriatric syndromes are an important factor influencing the quality of life in the older adult population. These syndromes are multifactorial, causing higher morbidity and impaired treatment outcomes [7]. Polymorbidity and geriatric syndromes are closely interconnected, and the occurrence of only two chronic diseases can lead to the development of geriatric syndromes, followed by the development of disability [8]. The main geriatric syndromes include falls, urinary incontinence, pressure ulcers, delirium, and functional decrease. The age, cognitive disorders, functional disorders, and immobility are perceived as risk factors [9]. Authors point out the occurrence of new geriatric syndromes, particularly frailty, sarcopenia, anorexia caused by ageing, and cognitive disorders. These syndromes are perceived as a presage of familiar syndromes, such as falls, depression, delirium, or hip fractures, and attention needs to be paid to the necessity of assessing new syndromes and adapting interventions in order to prevent the general acceleration of ageing [10]. Frailty, sarcopenia, loss of the body weight, and cognitive disorders have a high prevalence in the older adult population. The study confirmed the increasing prevalence of these syndromes with age; these syndromes occur most frequently in individuals above 85 [11]. The results also show an interconnection: 60% of frail individuals also suffered from sarcopenia, while, on the other hand, 40% of individuals with sarcopenia are frail at the same time. Highly qualified care involves the use of assessment tools since the care of the older adult population is rather complicated and demands a complex and multidisciplinary approach [12]. This fact suggests that the use of measuring and assessment tools will be a crucial factor for providing a high quality of care. The assessment tools that are used for the older adult population are specific and can determine the older adult’s state in several dimensions, i.e., health-related, cognitive, affective, social, spiritual, environmental, and functional states [13]. This study was mainly designed to draw the attention to the problem of geriatric syndromes of the Czech population above 60, and to the impact on the quality of their lives.

## 2. Materials and Methods

### 2.1. Aim and Study Design

The goal of the study was to show the presence of essential geriatric syndromes and their influence on the quality of life. This is a descriptive and explorative study focused on the 60+ population in hospitals or social care facilities, and in their home settings in the South Bohemian Region.

### 2.2. Study Participants

The study was performed in the South Bohemian Region, in the Czech Republic. The lower age limit was determined to be 60 years of age. The study was performed in hospital facilities, particularly in the aftercare departments, in the home setting, in elderly homes, or social care homes from May 2019 to April 2020. The participants were chosen using the method of a random choice according to quotas. Participants had to be more than 60 years old, live in the South Bohemian Region, and receive nursing care in the home setting or in selected hospital units. The size of the sample was determined by a statistician and based on the age of 60+ and the number of hospitalized individuals of this age and the number of recipients of home care. A total of 510 questionnaires were distributed; 502 filled-out questionnaires were returned. However, four of the returned questionnaires were excluded since they had not been filled out properly. The statistical analysis therefore comprises a sample of 498 respondents.

### 2.3. Assessment Tools

The data were collected using standardized and non-standardized questionnaires. The standardized part consisted of the Rapid Geriatric Assessment (RGA) questionnaire, which was, with the author’s consent, translated into Czech by two translators independently; the versions were verified, and the final version of the questionnaire was developed. For the assessment of the understandability of the questionnaires, a scale was created and tested in a pilot study. Within RGAS, the Fatigue, Resistance, Ambulation, Illness and Loss of Weight (FRAIL) questionnaire is determined to assess the frailty, strength, assistance with walking, rise from a chair, climb stairs and falls (SARC-F) is used to assess sarcopenia, the Simplified Nutritional Appetite Questionnaire (SNAQ) is used to assess nutrition, and the Rapid Cognitive Screen (RSC) is used to assess cognition. The quality of life in the older adults was researched using the WHOQOL-BREF assessment, which produces four domains of the quality of life: physical health, psychological, social relationships, and environment. 

The standardized questionnaire examines the ways in which a person assesses his/her overall quality of life, health, and wellbeing. For example, it describes the respondents’ characteristics, their independence, a subjective assessment of their physical and mental health, and a subjective assessment of the healthcare provided in the area of basic needs of daily living. Prior to the current data collection, a pilot study was conducted (January and February 2019) to assess the comprehensibility of the questions for the sample as well.

#### 2.3.1. Assessment of Sarcopenia

SARC-F is common tool to assess sarcopenia and is focused on the assessment of lifting and carrying of a 5 kg load, walking about a room, standing up from a chair, climbing 10 stairs or to the first floor, and falls during the last year [14]. The total score ranges between 0 and 10 points, and the value of 4 points was determined as a limit predicting sarcopenia and suggesting a poor result [15].

#### 2.3.2. Assessment of Frailty

FRAIL is a five-item tool focused on monitoring tiredness, resistance, exercise (movement), and loss of weight. The range is 0–5 points; 3–5 points represent the state of frailty, 1–2 points represent pre-frailty, and 0 points is no frailty [16]. The questions are focused on tiredness, climbing the stairs, walking 200 m, presence of more than five diseases, and a loss of weight during the last 6 months [17].

#### 2.3.3. Assessment of the Nutritional State

SNAQ is a simple four-item tool focused on the assessment of the appetite, the feeling of satiation after consuming a certain amount of food, enjoying the food, and the number of portions. Each question can be evaluated at 1–5 points. A score lower than 14 suggests a significant risk of 5% decrease of body weight in the last 6 months [18].

#### 2.3.4. Assessment of Cognition

The RCS is a tool used for a quick assessment of cognition and has been developed for the needs of primary care. The questions are focused on remembering five words, drawing a clock face and marking a particular time, answering a question based on a short story. The total score ranges between 0 and 10 points; 8–10 points represent normal state of cognition, 6–7 points represent a slight cognitive disorder, and 0–5 points represent a dementia [19].

#### 2.3.5. Assessment of Quality of Life

The WHOQOL-BREF is a self-administered questionnaire containing a total of 26 questions and is based on a four-domain structure of quality of life: physical health, psychological, social relationships, and environment. In addition, there are two included items from the Overall Quality of Life and General Health facet in the assessment that are examined separately. Each question can be answered using a five-item scale. Domains scores are scaled in a positive direction (i.e., higher scores denote higher quality of life). The mean score of items within each domain is used to calculate the domain score [20].

### 2.4. Ethics and Data Collection

The study was authorized by the Ethics Committee of the University of South Bohemia in 2019. The study was performed in accordance with the General Data Protection Regulation (GDPR) and in accordance with the Helsinki Declaration [21]. All participants were informed on the goals and sense of the study. Respondents were informed that they would agree with the data processing by filling out the questionnaire. Prior to filling out the questionnaire, detailed information was given on how to fill out the information.

### 2.5. Data Analysis

The obtained data were processed using the SASD and IBM SPSS 20 software. After data cleansing, a basic descriptive analysis of the file was processed, the descriptive statistics of the file were examined, and the data were entered into the pivot tables of the first and second classifications. Additionally explored was the distribution (using the Kolmogorov–Smirnov normality test) of individual variables, which subsequently determined the nature of the tests used. Furthermore, differences in mean values for the perception of quality of life in the WHOQOL-BREF assessment domains with respect to all of the abovementioned categories of standard questionnaires (SARC-F, FRAIL, SNAQ, and RCS) were tested using nonparametric tests (Mann–Whitney U test and Kruskal–Wallis test); all at the 5% level of significance. Then, the substantive significance of these results (effect size) was determined for the identified statistical differences, always with respect to the nature of the tests (Cohen’s d and Fisher’s eta^2^ - ŋ^2^).

## 3. Results

As part of the processing of the WHOQOL-BREF questionnaire, individual domain scores were first assessed; basic descriptive characteristics of individual domains were calculated (mean, standard deviation, maximum, and minimum). These statistics were examined in the domains of physical health, psychological, social relations, and the environment (see Table 1).

Furthermore, frailty was assessed using the FRAIL test. Out of the total (*n* = 498), 86 respondents (17.3%) were assessed as being without frailty, 277 respondents (55.6 %) were at risk of frailty, and 135 respondents (27.1 %) were found to be frail. A statistically significant difference (Mann–Whitney U test) in quality of life measured by WHOQOL-BREF was found between frail patients (according to FRAIL) and those without difficulties, especially in the domain of physical health and psychological, where large substantive significance was measured, which was tested using Cohen’s d; frail patients have significantly lower quality in both of these domains (see Table 2). The mean values of the quality of life in the domain of physical health were 12.82 and 8.59 in patients without frailty and in patients with frailty, respectively. In the domain of psychological, the mean values of the quality of life were 13.44 and 10.66 in patients without frailty and in patients with frailty, respectively.

Using SARC-F, sarcopenia was identified in 252 respondents (50.6%), and in 246 respondents (49.4%), sarcopenia was not confirmed through the SARC-F. As with fragility, a statistically significant difference in the quality of life measured by WHOQOL-BREF was also found between respondents with sarcopenia (decrease of muscle mass) (measured by SARC-F) and those without sarcopenia in all domains. However, in the domain of physical health and psychological, this difference was also substantially significant with a large size effect, tested using Cohen’s d, while in the other two, this effect was of medium significance. Patients with sarcopenia achieved significantly lower values. In respondents with no identified sarcopenia, the mean value of the quality of life in the domain of physical health was 13.85, in contrast to respondents with sarcopenia, where the mean value was 9.55. In the domain of psychological, the mean value of respondents without sarcopenia was 13.94, while it was 11.46 in respondents with sarcopenia. A lower difference was found in the domain of social relationships, where respondents without sarcopenia and respondents with sarcopenia achieved the mean values of 13.93 and 12.47, respectively. In the domain of environment, a difference in the quality of life was also observed: the mean value was 14.55 in respondents with sarcopenia, and 12.99 in respondents with sarcopenia. From the point of view of the effect size, high substantive significance was particularly identified in the quality of life of patients with sarcopenia, which was worse in the domains of physical health and psychological (with a high effect), and also in the domains of social relationships and environment (with medium effect) (Table 3).

The patient’s nutritional state was another area assessed using the RGA tool. The results were divided into two categories, as determined in the SNAQ tool, i.e.: 0—no risk, and 1—high risk of at least 5% of weight loss in the last six months. A total of 225 patients (45.2%) from the total number of 498 patients were assessed as without risk, and 273 patients (54.8%) came under the category of high risk. Differences in subjective perceptions of quality of life with respect to the categories of risk of weight loss (SNAQ questionnaire) were tested using the nonparametric Mann–Whitney U test. As with the two previous measurements, a statistically significant difference in quality of life measured by WHOQOL-BREF was again found between patients due to weight loss (according to SNAQ) and those without this risk in all domains. However, in the domain of physical health and experience, this difference was also materially significant with a large effect, tested using Cohen’s d, while in the other two this effect was of medium significance. Those with a high risk of at least 5% experienced weight loss. In the domain of physical health, the mean values of the quality of life were 13.42 and 10.23 in patients with no risk and patients at a high risk of losing at least 5% of weight, respectively. In the domain of psychological, the mean values of the quality of life were 13.99 and 11.61 in patients with no risk and patients at a high risk, respectively. The domain of social relationships had the mean value of 14.17 in patients with no risk, and 13.04 in patients at a high risk. Differences can be observed in the perception of the quality of life between the domains without risk and at a risk of losing weight. From the point of view of the effect size, particularly in the domains of physical health and psychological, there was high substantive significance (with respect to Cohen’s d). Unambiguously, the higher the risk of weight loss, the lower the quality of life, particularly in the domains of physical health and psychological (with high effect), but also in the domains of social relationships and environment (with medium effect) (Table 4).

Cognition was the last domain assessed using the RGA tool. Differences in subjective perceptions of quality of life with regards to the categories of level of cognitive impairment (RCS questionnaire) were tested using the nonparametric Kruskal–Wallis test. According to the RCS, three categories were determined: 0—normal cognition, 1—slight cognitive impairment, and 2—dementia. The total number of 498 patients consisted of 216 (43.4%) respondents with normal cognition, 195 (39.2%) with slightly impaired cognition, and 87 (17.5%) respondents were assessed as demented. Similarly, a statistically significant difference in quality of life measured by WHOQOL-BREF was found between patients with dementia and mild cognitive impairment (according to RCS) and those without problems, especially in the domain of physical health and psychological, where substantive significance was tested by Fisher’s eta^2^. The worse the measured cognition, the worse the perceived quality of life. In the domain of physical health, the mean value in the categories of normal cognition, slight cognitive impairment, and dementia were 12.78, 11.04, and 10.12, respectively. The mean values in the domain of psychological were 13.70, 12.28, and 11.09 for the categories of normal cognition, slight cognitive impairment, and dementia, respectively. In the domain of social relationships, the mean values were 13.75, 12.83, and 12.60 for the categories of normal cognition, slight cognitive impairment, and dementia, respectively. The domain of the environment had mean values of 14.18, 13.57, and 13.15 for the categories of normal cognition, slight cognitive impairment, and dementia. From the point of view of the effect size, for the domains of physical health and psychological, medium substantive significance was identified (according to Fischer’s eta^2^ (η^2^)). The higher the identified cognition, the lower the quality of life, particularly in the domains of the physical health and psychological (with medium effect), but also in the domains of social relationships and environment (with low effect) (Table 5).

## 4. Discussion

The study results showed that frailty, sarcopenia, nutritional state, and cognitive state had an impact on the perception of the quality of life. These are geriatric syndromes that, according to Sanford and collective [22], achieve a high prevalence across all kinds of care provided and have a negative impact on the quality of life, occurrence of disability, and elevation of mortality, though they are often neglected.

As far as frailty and the quality of life are concerned, the study by Lenardt and collective [23] showed that the higher the level of frailty was, the lower the level of the quality of life was. According to their study, frailty had the highest impact on the physical dimensions of the quality of life, in contrast to psychosocial dimensions where the impact was lower. The results in our study were similar, though frailty had a significant influence even on the domain of emotional experiences. The domain of social relationships was, similar to the study by Lenardt and collective [23], under a lower influence of frailty. The impact of frailty on the quality of life is also confirmed by Siriwardhana and collective [24], the results of which show a decrease by 7.3% and by 2.1% of the total score of the quality of life in association with frailty and the pre-frailty states, respectively. The impact of frailty on the quality of life was observed in all areas except for social relationships, home and neighborhood, and financial situation. 

According to our results, sarcopenia also has a significant influence on the quality of life. Sinesio Silva Neto and collective [25] came to the same conclusion, finding that sarcopenia had a negative impact on the quality of life. Using the SF 36 questionnaire, the highest impact was identified in the domains of physical limitations, physical pain, social functioning, and emotional problems. For the domain of the quality of life and sarcopenia, a specific tool named SarQol was developed. The SarQol is understandable for the older adult population and could be widely utilized for this domain in future. The authors state that the commonly used tools for the monitoring of the quality of life do not always identify details that can be caused by sarcopenia. However, some testing is still needed for SarQol questionnaires [26].

The risk of body weight loss, measured by the SNAQ, has, according to our study, a negative impact on the quality of life. The correlations between the nutritional state and the quality of life were studied by Luger and collective [27]. According to their study, nutritional state has a considerable influence on the quality of life, particularly in the domains of autonomy and social participation. Nutritional state and individuals at a risk of body weight loss in elderly homes were studied by Acar Tek and collective [28]. Using the SNAQ, 28.7% of respondents were assessed as at-risk individuals, and the level of the quality of life was significantly lower in women than in men. Furthermore, the attention was paid to obesity and the use of multiple medicaments, which can also significantly decrease the quality of life.

Our results show that the domain of cognitive functions has, similar to the abovementioned syndromes, an influence on the quality of life. Saracli and collective [29] perceive cognitive dysfunctions as a frequent cause of the loss of independence in older adult individuals, which has, according to the achieved results, a negative impact on the quality of life. The mental state is also negatively influenced by the depressive syndrome and other psychiatric disorders. Abrahamson and collective [30] came to the conclusion that the higher the level of cognitive disorders was, the lower the quality of life was, particularly in the domains of privacy, individuality, relationships, and mood. Arneson and collective [31] identified no significant correlation between the quality of life and a cognitive disorder, since, if appropriate interventions are applied, the tiredness and psychological distress will be reduced in individuals with a cognitive disorder.

## 5. Conclusions

The goal of the study was to show the presence of frailty, sarcopenia, mild cognitive disorder, and nutritional status and their impact on the quality of life. The results of the study proved the presence of frailty, sarcopenia, mild cognitive disorders, and a decreased nutritional state in the South Bohemian population older than 60 years. Furthermore, the results showed the impact of geriatric syndromes mainly on the domain of physical health and the psychological domain. In the nursing setting, frailty and sarcopenia need to be focused on in this population, since, currently, these problems are not monitored, and no sufficient nursing care is provided in this area. This is an essential finding, since interventions and further assessments should be adopted as a result of these changes. Currently, nutritional state, exercise, and cognitive training need to be monitored and adopted, which can have a preventive effect.

## 6. Limitations of the Study

The limitations of the study particularly result from the tools used and the setting where they were applied. Currently, the states are not assessed on a standard basis, and, therefore, there were no possibilities of a comparison except for our own findings. Further limitations can be caused by the nurses’ abilities to use tools which they commonly do not come across.

## Figures and Tables

**Table 1 healthcare-09-00657-t001:** Descriptions of domains scores of quality of life (WHOQOL-BREF).

Domains	*n*	Minimum	Maximum	Mean	St. Deviation
Physical health	498	4.00	20.00	11.67	3.29
Psychological	498	4.67	19.33	12.69	2.82
Social relationships	498	4.00	20.00	13.19	2.74
Environment	498	4.00	20.00	13.76	2.29
Valid *n*	498				

**Table 2 healthcare-09-00657-t002:** Assessment of the WHOQOL-BREF domains quality of life scores with respect to frailty (FRAIL) of older adults.

	Physical Health	Emotional Experiences	Social Relationships	Environment
Mann–Whitney U	6664.000	11,039.000	19,988.000	178,666.000
Wilcoxon W	15,844.000	20,219.000	29,168.000	27,046.000
Z	−12.515	−9.456	−3.207	−4.659
Asymp. Sig. (2-tailed)	*p* < 0.001	*p* < 0.001	*p* < 0.001	*p* < 0.001
Cohen´s d	1.12	0.85	0.29	0.42
Effect size	High	High	Low	Low

**Table 3 healthcare-09-00657-t003:** Assessment of the WHOQOL-BREF domains quality of life scores with respect to sarcopenia (SARC-F) in older adults.

	Physical Health	Emotional Experiences	Social Relationships	Environment
Mann–Whitney U	6779.500	15,296.500	21,500.500	18,395.000
Z	−15.105	−9.803	−5.997	−7.865
Asymp. Sig. (2-tailed)	*p* < 0.001	*p* < 0.001	*p* < 0.001	*p* < 0.001
Cohen´s d	1.35	0.88	0.54	0.70
Effect size	High	High	Mean	Mean

**Table 4 healthcare-09-00657-t004:** Assessment of the WHOQOL-BREF domains quality of life scores with respect to the quality of nutrition (SNAQ) in older adults.

	Physical Health	Emotional Experiences	Social Relationships	Environment
Mann–Whitney U	13,715.500	1593.000	19,419.000	18,305.000
Z	−10.651	−9.270	−7.165	−7.780
Asymp. Sig. (2-tailed)	*p* < 0.001	*p* < 0.001	*p* < 0.001	*p* < 0.001
Cohen´s d	0.95	0.83	0.64	0.70
Effect size	High	High	Mean	Mean

**Table 5 healthcare-09-00657-t005:** Assessment of the WHOQOL-BREF domains quality of life scores with respect to the level of cognition impairment (RCS) in older adults.

	Physical Health	Psychological	Social Relationships	Environment
Chi-square	58.313	54.692	14.870	14.800
df	2	2	2	2
Asymp. Sig.	*p* < 0.001	*p* < 0.001	*p* < 0.001	*p* < 0.001
Fishers eta^2^ (ƞ^2^)	0.12	0.11	0.03	0.03
Effect size	Medium	Medium	Low	Low

## Data Availability

The results of the research are available at the department of Science and Research of the Faculty of Health and Social Sciences, University of South Bohemia.

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
