# Peer review of "Impact of Selected Geriatric Syndromes on the Quality of Life in the Population Aged 60 and Older"

_healthcare, 2021, doi:10.3390/healthcare9060657_

Round 1
Reviewer 1 Report
Abstract:
Explain the abbreviation in the abstract. RGA.
Include p-values or comparison performed. Do not limited to state that all these variables affect all domain of quality of life. Provide further details in a statistical way.
Introduction
Please introduce the abbreviation WHO. Revise this fact in the manuscript.
Correct lines 38 and line 43. Sentences should start with a capital letter.
Correct Line 49. Reference is not well presented in this way.
Include the aim of the study in the introduction as well as further rationale of your study.
Statistical analysis and results
Authors state that "The correlations were tested using the Mann-Whit- 183
ney U test and Wilcoxon W tests". These tests do not show correlations, instead compare two groups (un-paired and paired respectively). Authors should revise the statistics and if they want to explore correlation, others tests are more appropiated.
Discussion
Include the limitations in this section.
Conclusion
Please be more concise.
Author Response
Based on your instructions, I made changes to the article, which I describe below exactly according to the points of your review. Thank you very much for the comment and I apologize for the mistakes I made in the original version.
Abstract: I put RGA into the abstract and provided the wording of the abbreviation. I entered the values ​​of p into the tables and I also specified the static evaluation in more detail.
I introduced the abbreviation WHO and others in the text. I correct al the references in the text. I try to explained the aim of the study in introduction.
Statistical analysis and results: After a discussion with the statistician, we came to the conclusion that this was not a statistical error. This is a problem of linguistics, and in that section I have modified the wording so that it already makes sense and is appropriate.
Conclusion: Based on this comment, I have modified the wording of the conclusion.
Reviewer 2 Report
- This manuscript appears to be an observational study utilizing screening tools apply to inpatient and home care patients to assess for different geriatric syndromes.
- Please use age friendly language throughout the manuscript - replace "elderly" with "older adults"
- Do not begin sentences with numbers, i.e. lines 38, 43, etc.
- Cite authors name with reference number, i.e. lines 228, 231, etc.
- For tables, re: significance, use p<.001
- Line 116 - identify the local ethics committee by name
- Line 81 - spell-out full names on first use of abbreviation Rapid Cognitive Screen (RSC)
- Line 23 - Rapid Cognitive Screen should be spelled out in the keywords
- Consider rephrasing Abstract lines 13-22: ... Cognitive impairment which are under-assessed geriatric syndromes. Methods: We used the Rapid Geriatric Assessment questionnaire to survey 498 hospitalized and home care adults aged 60 and older. Results: Frailty, ... Conclusions: Geriatric syndromes have an impact on quality of life, and incorporating assessments for frailty, sarcopenia, nutritional status, and cognitive impairment may enhance the care of older adults.
- Line 68: How were patients selected for the questionnaire? Convenience sampling, all admissions? Clarify the time frame - maybe January 2019-April 2020?
- Conclusions - consider rephrasing lines 270-277: The goal of the study was to identify selected geriatric syndromes...of the older adult population. Geriatric Nursing can positively impact care and quality of life for older adults assessment of frailty, sarcopenia, nutritional status, and cognitive impairment.
- Conclusions - consider rephrasing lines 281-285: ...of these changes. In the future, nursing will encounter more older adults particularly in the rapidly expanding areas of home care telemedicine.
- Limitations - consider rephrasing lines 287-291: This investigation was an observational study utilizing screening tools applied to inpatient and home care patients. This study was also limited to a part of the Czech Republic....
Author Response
Based on your instructions, I made changes to the article, which I describe below exactly according to the points of your review. Thank you very much for the comment and I apologize for the mistakes I made in the original version.
2. I replaced "elderly" with "older adults" in all the text.
3. I replaced the number of the citation on the end of the sentences.
4. I cited authors name with reference number.
5. In the tables I used significance p<.001.
6. I complete the name of the local committee into the text.
7. I spell-out full names on first use of abbreviation when using all abbreviations for the first time.
8. I put all the parts of RGA into the key words.
9. Based on this comment, I reformulated the wording of the abstract.
10. I comlete these information into the text.
11. Based on this comment, I reformulated the wording of the conclusion.
13. I removed the sentence, leaving the rest of the limitations.
Round 2
Reviewer 1 Report
I reiterate my concern about statistics. I think the approach is interesting but that the statistical tests chosen are either not the most correct or not adequately explained.I think that in the statistics section it should be reflected that some subgroups will be created based on the score of the questionnaires as well as explaining what the cut-off points of these selected questionnaires will be. Then the authors must choose whether to perform correlations or independent sample t-tests. I think the two concepts are mixed and this should be clear. In this regard, tables´ title should be corrected according to the statistical test performed.
Author Response
In the following revision, we made major changes.They cover almost all main areas (Abstract, study pasrticipants, assessment tools, data analysis, results and cocnlusion).
This version should be clearer and easier to understand.
We focused very much on clearly describing the statistics and made clearer explanations.
Thank you for your comments.
Reviewer 2 Report
- The authors have made a credible response to the reviewer comments but the abstract, methods, and conclusion remain unclear. Consider merging the content from the original abstract, especially the study numbers, and information contained in the original conclusion with the revised sections.
- Abstract: line 11, replace"hot topic" with a "critical domain"
line14, … assess geriatric syndromes...
lines 15-17… Non parametric testing revealed... - Participants, line 73 -provide more detail on the randomization such as subsequent discharges or a convenience sample performed within a specified time.
Author Response
In the following revision, we made major changes.They cover almost all main areas (Abstract, study pasrticipants, assessment tools, data analysis, results and cocnlusion).
This version should be clearer and easier to understand.
Thank you for your comments.